# The “www” of *Xenopus laevis* Oocytes: The Why, When, What of *Xenopus laevis* Oocytes in Membrane Transporters Research

**DOI:** 10.3390/membranes12100927

**Published:** 2022-09-25

**Authors:** Manan Bhatt, Angela Di Iacovo, Tiziana Romanazzi, Cristina Roseti, Raffaella Cinquetti, Elena Bossi

**Affiliations:** 1Laboratory of Cellular and Molecular Physiology, Department of Biotechnology and Life Sciences, University of Insubria, Via Dunant 3, 21100 Varese, Italy; 2Experimental and Translational Medicine, University of Insubria, Via Ottorino Rossi 9, 21100 Varese, Italy; 3Centre for Neuroscience—Via Manara 7, University of Insubria, 21052 Busto Arsizio, Italy

**Keywords:** *Xenopus* oocyte, solute carrier, two-electrode voltage clamp, fluorophores, RNA microinjection, membrane transplantation, membrane transporter, immunochemistry, single-oocyte chemiluminescence

## Abstract

After 50 years, the heterologous expression of proteins in *Xenopus laevis* oocytes is still essential in many research fields. New approaches and revised protocols, but also classical methods, such as the two-electrode voltage clamp, are applied in studying membrane transporters. New and old methods for investigating the activity and the expression of Solute Carriers (SLC) are reviewed, and the kinds of experiment that are still useful to perform with this kind of cell are reported. *Xenopus laevis* oocytes at the full-grown stage have a highly efficient biosynthetic apparatus that correctly targets functional proteins at the defined compartment. This small protein factory can produce, fold, and localize almost any kind of wild-type or recombinant protein; some tricks are required to obtain high expression and to verify the functionality. The methodologies examined here are mainly related to research in the field of membrane transporters. This work is certainly not exhaustive; it has been carried out to be helpful to researchers who want to quickly find suggestions and detailed indications when investigating the functionality and expression of the different members of the solute carrier families.

## 1. Introduction

Fifty years after the first heterologous expression of a protein in *Xenopus laevis*, it is necessary to wonder why it is still used in membrane protein studies, when it is the appropriate model, and, finally, what kinds of experiments are worthwhile to perform with these cells. *Xenopus laevis* oocytes have been used for a long time in the study of membrane proteins, particularly in the functional characterization of transporters and channels, and they are specific model systems. *Xenopus laevis* oocytes are fully equipped with translational machinery [1,2,3] as the full-grown oocyte is a repository of maternal mRNAs and proteins ready for early embryogenesis; these tools can be exploited to translate and correctly localize exogenous, microinjected RNAs. Consequently, the expression of heterologous proteins is usually very high with a very low background signal. A great number of endogenous electrogenic proteins and membrane conductance are characterized and known [4,5,6,7,8,9,10,11,12,13,14]. The multimeric proteins are correctly assembled (more details are reported in these references [15,16,17,18])**,** and the biophysical parameters are comparable to those collected with different approaches in the primary culture [19], cell line [20,21], and native membrane environment [22]. Moreover, they are giant cells; a mature oocyte has a diameter of about 1–1.2 mm; and they can be easily prepared, injected, and tested with different approaches. In this review, different examples of applications, tricks, and protocols useful for characterizing membrane protein are reported (Figure 1) with the goal of suggesting new and old methods to solve problems in membrane transporter studies, showing how *Xenopus* oocytes are still a formidable tool for studying a large variety of significant biological questions.

## 2. *Xenopus laevis* Oocytes as a Heterologous System of Expression

### 2.1. mRNA Injection

In the early 1970s, Gurdon [23,24,25,26] reported the capability of amphibian oocytes to translate foreign mRNA. Ten years later, Sumikawa [27] and Miledi [28,29,30] expressed and studied functional membrane proteins.

*Xenopus laevis* oocytes have a highly efficient biosynthetic apparatus that performs all the post-translational modifications needed for correct protein targeting and function; it is possible to co-express different proteins or subunits by co-injections of the corresponding mRNAs [31]. After 48–72 h, if the cRNA is transcribed from a suitable plasmid construct (see the indication in Appendix A), the proteins are translated into exogenous proteins ready to be tested for their functionality. It is necessary to remember that the expression is transient, according to the stability of the mRNA and the half-life of the protein; it lasts for two to seven days. In the culture medium, the selected and the injected oocytes can stay healthy for many days. They can be used with the method described here for about 10 days from the collection day.

### 2.2. Membrane Microtransplantation from Native Tissues

The possibility of studying the properties of heterologous proteins is not limited to mRNA injection. An effective tool for studying exogenous proteins is also offered by the technique of microtransplantation of receptors and channels from native tissues into Xenopus laevis oocytes.

Although the oocyte represents a powerful approach to studying different membrane proteins and their functional and pharmacological properties, it is not able to completely reflect the native environment, which could confer modifications in the functionality of membrane proteins compared to that seen in oocytes. Interestingly, Xenopus laevis oocytes can be used to overcome this limitation through the technique of microtransplantation of membrane patches from human (or from other species) tissues [32,33]. This technique allows the study of the original membrane proteins and the associated molecules, as interaction or anchoring proteins are still embedded in their natural lipid environment. Indeed, it is possible to investigate the neurotransmitter receptors and ion channels that maintain all their native properties. Starting from a very small amount of tissue, the great advantage of microtransplantation in Xenopus laevis oocytes is that it enables the use of human samples (from autopsies or patients who have undergone surgery) to perform functional studies. This approach has been revealed to be widely useful in many fields, allowing the characterization of the role of membrane proteins in different human diseases, from neurodevelopment disorders to neurodegeneration pathology, epilepsy, and channelopathies [34,35,36,37,38,39]. Particularly, membranes tissue transplantation allows the investigation of the role of neurotransmitters, such as GABA, glutamate, and acetylcholine, in human brain diseases where impairment of neurotransmitters or their cognate receptors or altered levels of ion concentration can induce brain damage. This technique can also be applied for the screening of novel drug compounds directly on active membrane proteins as hypothetical targets [40,41]. Oocytes microtransplanted with membrane fragments can help us to understand the mechanisms of action of substances (i.e., inflammatory mediators) on the ion channels present in the original tissue [38,42]. Using the transplantation of cortical membranes from Temporal Lobe Epilepsy (TLE) patients in Xenopus laevis oocytes, the role of EPO in improving the functionality of the GABAA receptors was demonstrated [43]. GABAergic and glutamate transmissions have been studied in Rett syndrome through the microtransplantation method, comparing the electrophysiological results obtained from injected tissues from the human and murine cortexes in Xenopus laevis oocytes [44]. Via microtransplantation of human muscle membranes, Palma et al. studied functional acetylcholine receptors (AChRs) isolated from muscle biopsies of patients with Amyotrophic Lateral Sclerosis (ALS), assuming that AChR is a useful target for ALS pharmacological therapy [45]. The above examples highlight the ability of oocytes to enable the study of specific human pathologies from small amounts of pathological tissue without any kind of alteration induced by endogenous cell machinery. A single oocyte can integrate the transplanted membrane fragments into its membrane, allowing for the direct study of ion channels or receptors in their native environment (e.g., by TEVC). Details about membrane preparation and microinjection are reported in Appendix A.

## 3. Testing the Function of Membrane Proteins

Once the proteins are correctly expressed, there are many different approaches to test their functionality. Descriptions of some of the methods are reported in this paragraph with relatively useful literature suggesting when the *Xenopus laevis* oocytes are a perfect tool and what results can be achieved using them by which experimental approach.

### 3.1. Classical TEVC

The functional study of protein expressed in Xenopus laevis oocytes historically uses electrophysiological recording by two-electrodes voltage clamp (TEVC), which is the standard technique due to the dimension of the oocytes. The possibility of controlling the membrane voltage allows the researcher to collect data about the ionic current in defined experimental conditions. Membrane voltage, external solution, substrates, and agonists can be easily applied and modified during the experiments, and it is also possible to inject drugs, chelators, substrates, and ions inside the cell during the experiments. Specifically, a TEVC experiment consists of the recording of the whole oocyte membrane current while applying membrane voltage changes and/or exchanging the extracellular solution through the perfusion system [46,47]. The change in the electrogenic ion fluxes across the membrane is measured as a variation of the current, i.e., equal in amplitude but opposite in sign, at the output of the amplifier connected to the current electrode. By convention, the influx of anions (or efflux of cations) is reported as an upward deflection (outward or positive current) and the influx of cations (efflux of anions) as a downward deflection (inward or negative current). Some proteins drive ions in precise directions (downward with the electrochemical gradients) as many ion channels, but some others can simultaneously transport two or three different types of ions that may have opposite charges or that are translocated in opposite directions [48] (also against their electrochemical gradient); the total ion charge across the membrane is due to a movement of different charges in a different direction.

Xenopus laevis oocytes in the 1990s had a very important role in membrane transporter characterization. After the first solute carriers were cloned [49,50], a great amount of RNA coding for similar proteins became available, and the main tool for their characterization was the expression cloning technique [51], which contributed to the knowledge of the substrates of a huge number of transporter proteins. In this paper [52], it is possible to find an update of the technique with a comprehensible protocol.

Solute carriers are still understudied [53,54,55], and their roles in physiology, pathology, drug targeting, or drug delivery are continuously growing in importance. Our experience in investigating these proteins is the basis of the experimental suggestion described in this review, with which we would like to help researchers to find the right methods to improve knowledge of SLC proteins. 

The possibility of expressing heterologous recombinant cDNA is fundamental not only to understanding the function of the protein but also to identifying the role of the different functional determinants such as single amino acids, the specific transmembrane domain, or internal or extracellular loops. There are hundreds of examples of mutants or chimeric proteins that anticipated the discovery of the localization or the function of the transmembrane domains or single determinants, confirmed years later by the crystal structure. Kanner [56,57], by cysteine scanning and mutagenesis, identified the possible role of the first transmembrane domain of the SLC6 family established by LeuT_Aa_ structure [58]. Given the possibility of creating recombinant proteins, another interesting approach is the construction of chimera proteins [59,60,61,62,63], for example, to identify the substrate and ions binding site. The same approaches can be also used to express a protein at the plasma membrane even if, in vivo, it is expressed in a different compartment; this can help to characterize the transport by TEVC in Xenopus laevis oocytes [64,65].

The possibility of changing specific residues can also modify the electrogenicity of the transporter [66] or explain the differences between orthologs; this approach is widely used to recognize the activity of mammalian transporters [67,68,69,70,71,72,73]. 

In the field of membrane transporters research, the possibility of working with mutants, in the beginning, was the basis for understanding the structure and functional relationship between the different domains of the proteins. After the crystal structures, single and multiple mutants became the tools for confirming the functional role of the regions identified by the solved structure [71,74,75,76,77,78,79,80,81,82].

Introducing one or multiple amino acid substitutions in membrane transporters is quite simple because it is possible to easily modify the cDNA sequences of the original transporters using an appropriate commercial kit or available protocols [83,84,85]. Once the cDNA has the desired mutations, it can be easily transcribed, and the corresponding cRNA and the mutated protein are produced by the oocytes. The importance of this approach is certified by thousands of publications. In many cases, the current elicited by the electrogenic transporter is large enough that possible reduction of the function of the mutated transporter can be still characterized as the currents are still easily measurable. 

Again, due to the size of the currents, it is also possible to appreciate changes in their shape; this is particularly true if pre-steady-state currents are investigated. These currents are slow, transient currents related to the presence of the transporter in the plasma membrane and are only observed in oocytes expressing membrane transporters that present electrogenic steps in their transport cycle (Figure 2). These currents are elicited by voltage jumps or concentration jumps applied to oocytes expressing the transporter of interest [86,87,88,89]. The recorded currents are visible in the absence (in the majority of the cases) or even in the presence of the substrate (in many cases for concentrations below the saturating concentration of the substrate [77,90,91]). The pre-steady-state currents are a tool for studying the interaction between ion(s) and substrates and the transport protein during the translocation cycle. Moreover, in some cases, they are also helpful for verifying the localization of specific residues (i.e., charged residues) in the membrane electric field [92,93,94,95,96] or to determine the apparent affinity of the transporter for a specific substrate [77].

For some transporters, the pre-steady-state currents are particularly large and represent a means for specific biophysical characterization, for example, GAT1 (slc6a1) [86,98], SGLUT (SLC2) [99], PepT1 and 2 (SLC15) [91,94,96,100], and NaPi (SLC34) [101]. Another interesting aspect of the heterologous expression of membrane transporter in Xenopus laevis oocytes is the possibility of testing the behavior of the same cell at different temperatures to estimate the steps of the transport cycle that require larger conformational changes. The larger variation of the temperature coefficients (Q_10_) for the different parameters that can be acquired is related to larger conformational changes involved in that specific function; the weaker temperature sensitivity is, instead, related to the diffusional component of the process [92,95,102,103,104].

Xenopus laevis oocytes have the capability of correctly expressing the different subunits that can be exploited to understand the role of every single subunit in the ion channels, but, also in membrane transporter research, the role of subunits can be investigated [105], even with a peculiar construct [106].

Xenopus laevis oocytes are also used to study reverse transport, i.e., by injecting the substrates. This approach can be particularly helpful for understanding the interaction with the different ions in the transport cycles [107,108]. Moreover, due to the dimension of the cells, it is also possible to evaluate the changes in the intracellular contents, i.e., pH changes [109]. The intracellular environment can be also changed by co-expressing ion exchangers or with specific incubation solutions [110]. The concentration of Cl^−^ can be estimated [108] using the presence of endogenous CaCC (Ca^2+^-gated Cl^−^ channel) [5]. This channel, constitutively present in mature oocytes, is one of the most widely used tools in the investigation of exogenous metabotropic receptors in Xenopus laevis oocytes. Since 1987, when the acetylcholine- and neurotensin-evoked responses of oocytes were studied, the activation of this channel by the DAG/IP3 signaling pathway has been used to characterize an impressive number of G-protein-coupled receptors that have been exhaustively characterized in the oocyte [111,112,113]. Similarly, Xenopus laevis oocytes are useful tools for studying “Store-Operated Ca2^+^Entry” (SOCE). In the paper of Courjaret [114], different methodologies to investigate this pathway are reported.

### 3.2. Voltage Clamp Fluorimetry (VCF)

Xenopus laevis oocytes can be utilized to investigate the conformational movements of single amino acid residues by monitoring the changes in fluorescence emitted by a fluorophore attached to the residues in the selected positions. With high temporal resolution, this functional investigation provides information on the conformational changes in a protein when its environment is changed in terms of substrates, voltages, pHs, etc., simultaneously allowing observation of its function (Figure 3). The power of functional, site-directed fluorometry is elegantly reported in [115]. The technique is applied to study the interactions of the electrogenic protein with modulators. An example of applications of VCF to membrane transporters is reported by Virkki et al. [116]; this paper details the set-up used. Oocytes and VCF have been also used to improve optical voltage indicators [117].

## 4. Testing the Expression of Membrane Proteins

### SOC and Immunochemistry

When mutants are expressed in Xenopus laevis oocytes and show loss or gain of function, revealed by the previously described technique, it is necessary to verify the possible alteration of the expression of the transporter at the level of the plasma membrane. The quantification of the protein can also be useful for investigating the effect of modulation or the co-expression of accessory subunits. The immunostaining of heterologous protein is a valid tool not only for detecting and quantifying the expression, but also for following the trafficking of the protein [120,121] or verifying the targeting on the plasma membrane [122], even if the correct targeting is, in many cases, verified by the function; when the function is hampered, confirmation of the presence or the absence of the protein at the plasma membrane becomes of great importance [123]. It is also useful to point out that the classical Western blot technique is not the first choice for investigating the presence of the heterologous protein at the plasma membrane, primarily because the high presence of lipoprotein in many cases masks the signals and because it requires the separation of the membranes [124] and/or the application of immunoprecipitation techniques [18,125], or the biotinylation of the membrane protein [72,106] Western blot from homogenized oocytes requires specific tricks to even verify the presence of cytosolic proteins [123,126]. To our knowledge, there are two other effective immunochemistry techniques for investigating exogenous proteins in Xenopus laevis oocytes which are easier and less time-consuming, even if they are not applicable for all the proteins expressed.

One of these approaches is Single-Oocyte Chemiluminescence (SOC), which allows a comparison of the amount of protein expressed on the oocyte membrane in different groups of differently treated oocytes or expressing mutants or accessory subunits. Zerangue et al. [127] are the pioneers of this approach. The SOC technique uses classical immunostaining on the whole oocyte labeled with a secondary antibody conjugated to Horseradish Peroxidase (HPR). Many oocytes can be tested at the same time, and the chemiluminescence can be detected in a 96-well plate using a luminometer. The oocytes are fixed in paraformaldehyde and incubated in the primary and in the secondary antibody peroxidase conjugate, allowing signal detection proportional to the amount of protein present in a single oocyte (see Appendix A for detailed protocol). The chemiluminescence emitted is quantified with a microplate reader. An advantage of this technique is that the detection on a multiwell plate and data collection from a single oocyte allows the analysis of many cells at the same time, providing statistically significant numbers in a shorter time than other techniques. SOC is a valid approach for comparing the amount of mutant protein expressed at the plasma membrane and verifying the relationship between the function and the amount of protein present in the plasma membrane. One example of this approach was the correlation between the function and the expression of some rbPepT1 mutants demonstrated by Bossi et al. This paper [122] reported the electrophysiological characterization of FLAG-rbPepT1 mutations in charged residues, with the example of non-functional mutants that, in one case, were correctly localized in the membrane, but, in another, the introduced mutations completely impaired the correct localization of the membrane. In this case, a modified protein with a FLAG sequence in the V extracellular loop was used [128]. The quantification of the expression by oocytes can also be used to evaluate the effect of the presence of the accessory protein, as reported for B0AT1 [105]. In our experience, this technique works when the antibody is against a sequence located in the loops spanning the extracellular side of the protein, possibly in extracellular loops. Epitopes located in the transmembrane domains or the intracellular loops are not (well) recognized. Moreover, cytoplasmatic proteins cannot be labeled because of the presence of the membrane that, even if partially permeabilized, limits the diffusion of the antibody inside the cell due to the dimension. The localization of the heterologous protein can also be performed by the immunostaining of the oocytes’ section. With this technique, cytoplasmatic proteins or internal epitopes can be well recognized. Oocytes, in this case, are fixed with 4% paraformaldehyde embedded into a cryo-embedding compound (Polyfreeze tissue-freezing medium), frozen in liquid nitrogen, and cut into thin (7 μm) sections with a cryostat. Then, the slices are incubated with the primary and secondary antibody fluorophore conjugate and visualized by a fluorescence microscope, as detailed in Appendix A.

## 5. Alternative Methods to Quantify the Functionality

There are experimental conditions where it is not possible to investigate the function of the protein expressed by applying classical approaches. In our research experience, we have developed, modified, or identified specific strategies to overcome the various obstacles that arise in peculiar investigations. To increase the power of Xenopus laevis oocytes as a research tool, we report some suggestions to better utilize this system.

### 5.1. Fluorescence Monitoring

Autofluorescent and opaque oocytes have been used on several occasions to monitor fluorescent molecules, both as substrates and as indicators [129,130,131] (Figure 4). The paper of Illing [130] is a milestone in using fluorescence for monitoring the functionality of membrane transporters. It provided validation for the use of PhenGreenSK fluorescence quenching as a marker of cellular metal ion uptake. In this paper, they determined the metal-ion selectivity of DMT1 (SLC11a2) in specific conditions using a voltage clamp, radiotracer, and fluorescence assays, validating the methods. For the characterization of the dictyostelium discoideum orthologs of DMT1, NRAMP1, and 2, calcein, a more common (and cheaper) fluorescence indicator, was used both in the cells of the amoeba and in Xenopus laevis oocytes [64]. In this work, the quenching of calcein was monitored by confocal microscopy on a single oocyte, but the technique also works with basic LED-based fluorescence microscopy with equal detection capacity, as reported in [132] and as shown in the various experiments carried out to validate the method and in the subsequent peculiar oocytes applications [133,134]. The method proposed by Illing and modified by our group works well, but it has two disadvantages. First, it requires a lot of time to test the number of samples to achieve statistical significance, considering that, according to the transporter efficiency/activity testing, the quenching of one oocyte requires between 5 and 30 min. The second problem is related to the autofluorescence of the oocytes, which changes from batch to batch, and, consequently, needs collecting for each experiment, as well as several reference samples. To solve the first problem, Cinquetti et al. [135] developed a method in which the quenching of the fluorophore was used to indirectly monitor the translocation of substrates or ions, using a plate reader to collect a satisfactory number of data at the same time. The technique reported in [135] can be applied to measure the difference of the fluorescence in a cytoplasmic extract from the oocytes exposed or not to the substrate and is able to quench the fluorophore, indirectly measuring the uptake. In the same paper, autofluorescence was also investigated in this kind of assay. 

Detailed protocols for the use of fluorescence indicators to monitor the changes in the content of the cytoplasmic environment, and, consequently, indirectly the oocytes uptake, can be found in the cited papers, and a good strategy to avoid the problem with autofluorescence is reported in [136].

### 5.2. Monitoring the Transport by HPLC, GC-MS 

One of the classic methods of screening and profiling compounds for a transporter is the radiolabeled cellular uptake [52]. This method provides solid evidence of the substrate or inhibitor characteristics of a compound, but it also produces unwanted radioactive waste while involving a time-consuming, costly procedure to radiolabel the compounds. Moreover, in recent years, many institutions have preferred radioactive-free experiments because working with radiolabeled compounds, when they exist, has high costs, high risks for the staff and the environment, and requires a specifically controlled and isolated working area. As an economical, accurate, and time-efficient method, high-performance liquid chromatography (HPLC) provides a very appealing alternative but is still underused. Since the 1950s, scientists have been optimizing different protocols for chromatographic separation using HPLC, but preparing the sample and choosing the right detector for analyzing the cytoplasmic content of the oocytes remains a challenge. Consequently, only a small number of papers have reported HPLC as a method for quantifying uptake in Xenopus laevis oocytes, even if it has been used since the 1990s [137]. 

The HPLC-based approach for Xenopus laevis oocytes expressing transporter provides a time- and cost-efficient tool for profiling known and unknown compounds [138]. This method, when combined with TEVC, can provide a correlation between the substrate and the charge translocation and allow discriminate between the real substrate and possible molecules acting as channel openers, as reported in [139]; in this paper, the D-serine was confirmed to be a substrate of KAAT1 (K^+^-coupled amino acid transporter 1).

### 5.3. High-Throughput Systems

Conventional electrophysiological experiments using a manual, two-electrode voltage clamp provide robust, high-resolution, and direct results for electrogenic protein activities but a low turnover. Given the sheer number of compounds to be tested and the unclassified transporters, ion channels, and receptors, it is essential to develop sensitive, automated, and optimized processes for direct electrophysiological measurements [140]. Several groups have developed high-throughput systems with automated recording stations for TEVCs. The Roboocyte™ (developed by MCS GmbH, 72770 Reutlingen-Germany) allows automated cDNA/cRNA microinjection into Xenopus laevis oocytes and TEVC recording from oocytes in a 96-well plate. Another system, named OpusXpress™ (developed by Axon Instruments, USA), can run experiments using eight recording stations. The Parallel Oocyte Electrophysiology Test stations (POETs™ developed by Abbot Laboratories, Abbot Park, USA) provide a throughput of ~14 plates/day (96-well) and are extensively used to study ligand-gated ion channels [141,142]. As technology grows, these companies also upgrade their high-throughput systems. For example, the Roboocyte™ was upgraded and launched as Roboocyte2™ with upgraded acquisition software, faster solution exchange, and a compact design with increased automation.

As a general concept, these automated devices can measure 96 oocytes in one go, using the standard, commercially available 96-well plates. The perfusion systems are controlled by a robot head, but scientists can also modify the system as needed. These systems have been used to study GABA receptors, Na^+^/K^+^-ATPase transporter, and various ion channels [140,142,143]. These automated high-throughput systems have had an impact, increasing the contribution of electrophysiology (using oocytes) for studying the effects of substrate compounds and discovering new molecules.

## 6. Conclusions

The techniques reported here are only a partial summary of the possible applications available that use Xenopus laevis oocytes in research. The methodologies examined are mainly related to research in the field of membrane transporters, which is our research activity. This work is certainly not exhaustive, but we hope it will be useful to researchers who want to quickly find suggestions and detailed indications when investigating the functionality and expression of the different members of the solute carrier families.

## Figures and Tables

**Figure 1 membranes-12-00927-f001:**
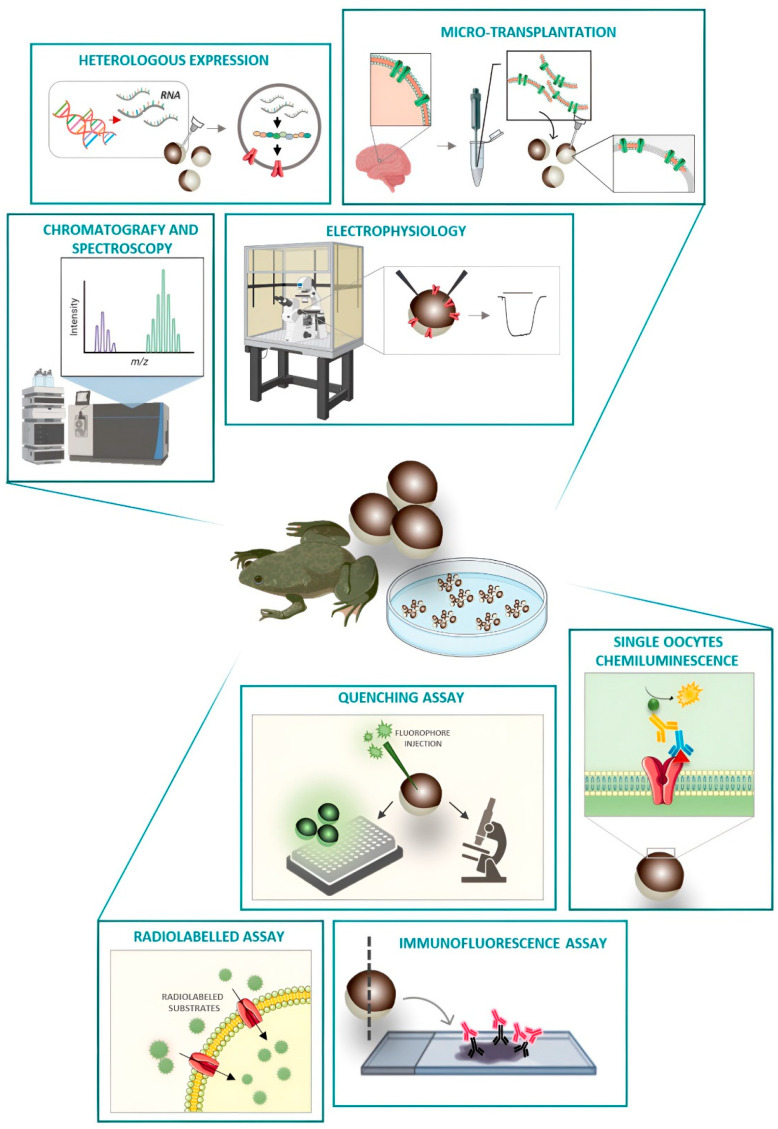
Schematic drawing of use and applications of Xenopus laevis oocytes. Each section represents a possible use of X. laevis oocytes described in this paper. The use of oocytes as a heterologous expression system allows amplification of the signal and characterization of defined membrane protein(s) starting from cRNA or the original membrane where the protein(s) was previously expressed. Isolated tissue from human or animal models can be microtransplanted in X. laevis oocytes studying a plethora of channels and receptors with native properties maintained in the native environment. Chromatography and spectrofluorimetric procedures can be performed to quantify the presence of substrate inside the Xenopus laevis oocytes. Electrogenic membrane proteins can be investigated via TEVC. X. laevis oocytes injected with cRNA of interest or transplanted with tissue samples can be also tested for function using radiolabelling or fluorimetric assay and for expression by immunochemistry techniques.

**Figure 2 membranes-12-00927-f002:**
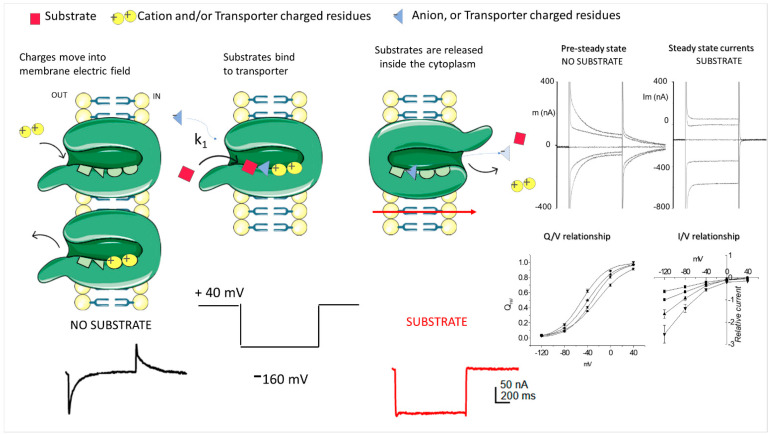
The cartoon modified from [87] represents a simple scheme of two of the many other states from the transport cycle: the outward open conformation ready to bind the substrate (left) with the substrates bound (center) and the inward open conformation with the released substrate. The pre-steady-state currents are visible when a rapid voltage change is applied. The ions are moved deeper into the transporter vestibule, or (and) charged residues(s) are rearranged. The charges moved in the membrane electric field change the amount of charge of the membrane capacitor and can be revealed as a slow relaxation of the oocyte capacitive current. These currents are isolated from the endogenous capacitive currents by the subtraction of the current recorded in the presence of an inhibitor (when available) or by a double exponential fitting [97], considering the fast component related to the capacitor behavior of the plasma membrane of the oocytes and the slow behavior to the presence in the membrane of the transporter. The pre-steady-state currents are transient; the area under the transient at the “on” of the voltage pulse is equal to that of the “off” and shows saturation at extreme positive and negative voltages [93]. In the absence of a substrate, the charges cannot move further, so, when the voltage comes back to the starting value, the charges move out, giving rise to the transient off. The integration of the area of the relaxation allows quantification of the number of the charges involved. When the substrate is present, it elicits conformational changes, completing the transport cycle to the inward open conformation (right) moving substrate and cotransporter ions inside the cell, generating the transport current. The inward open conformation is simply schematized by the opening of the gate in the presence of the substrates. On the right is an example of transient currents and transport currents and the data that can be collected from them regarding the Q/V (charge/voltage) and I/V (current/voltage) relationship.

**Figure 3 membranes-12-00927-f003:**
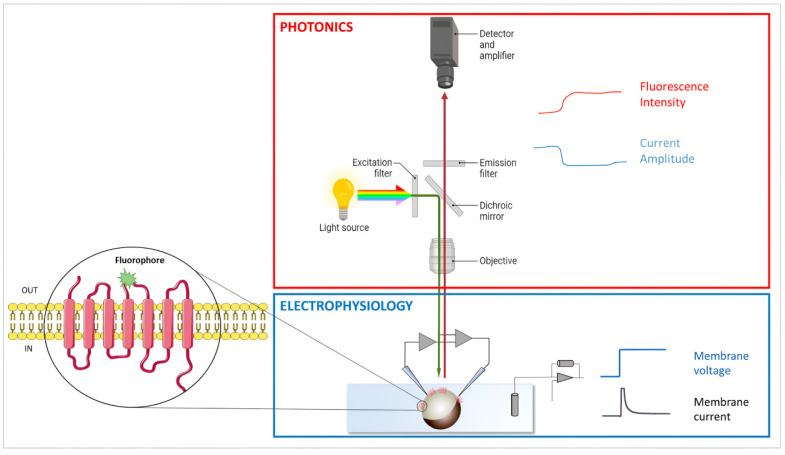
Identified residue is mutated in cysteine that can be bound to a fluorophore. The engineered cysteine residue with the tethered fluorophore is represented by the green star. Changes in membrane potential, pH, and concentration cause fluorophore displacement, changing the fluorophore environment and, consequently, fluorescence emission. Membrane current and fluorescence intensity can be simultaneously measured. A detailed explanation of the methods can be found here: [118,119].

**Figure 4 membranes-12-00927-f004:**
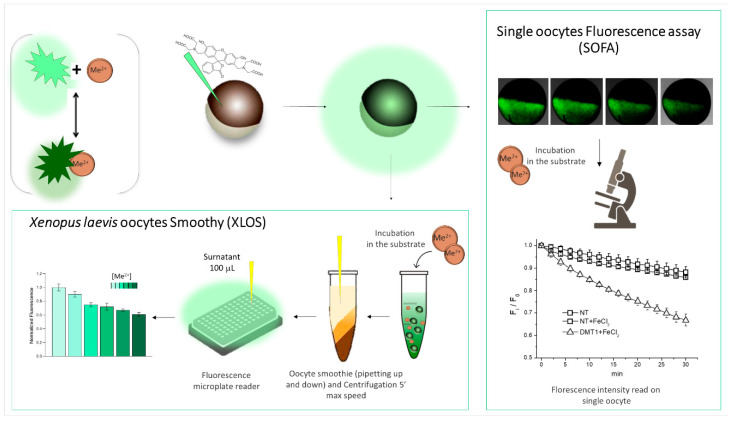
A fluorophore that reduces or increases its emission in the presence of the substrate can be used to monitor the changes inside the oocytes. Calcein reduces its emission intensity when interacting at physiological pH with divalent cations. Calcein-injected oocytes expressing the divalent metal transporter DMT1 (slc11a2) are incubated in a solution containing Me^2+^. It is possible to follow the fluorescence quenching by monitoring the decay of fluorescence intensity with a fluorescence microscope. It is also possible to measure the final fluorescence intensity in many oocytes at the same time, incubating them in solution with or without the Me^2+^, and measuring the fluorescence intensity of the supernatant obtained after the homogenization and centrifugation of the oocytes.

## Data Availability

This review do not report no unpublished data.

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
