# Peer review of "The “www” of Xenopus laevis Oocytes: The Why, When, What of Xenopus laevis Oocytes in Membrane Transporters Research"

_membranes, 2022, doi:10.3390/membranes12100927_

Round 1

Reviewer 1 Report

Manan et al provide a timely and succinct review of the use of Xenopus oocytes as a heterologous expression system to characterize the function on ion transporting proteins (SLC transporters).  The review is very well written and easy to read, which will likely make it a great resource for students and trainees new to the technique.  As someone who is very familiar with using Xenopus oocytes to characterize ion transporters I was pleased to learn about approaches that I was previously not familiar with, such as micro-transplantation and SOC.  Overall, this is a very nice contribution that will be of great use to the field of membrane biology.  Below are some minor suggestions to improve the manuscript:

1)      The addition of few figures showing representative data from published studies would be helpful for illuminating some of the techniques that are not that easy for a non-expert reader. For example, the description of pre-steady state currents and their interpretation (lines 203-212) are not easily understood.  A figure illustrating the details the authors are referring to along with some representative data would help better communicate this approach.   Figures would also be helpful for comprehension of the VCF technique (lines 231-240) and fluorescence monitoring (lines 299-328). 

2    A few minor spelling/grammatical errors:

a.         line 233, ‘resides’ should be ‘residues’;     

b.       Line 370,   ‘ATP’ should be ‘ATPase’

Author Response

Responses to the Reviewers

Manan et al provide a timely and succinct review of the use of Xenopus oocytes as a heterologous expression system to characterize the function on ion transporting proteins (SLC transporters).  The review is very well written and easy to read, which will likely make it a great resource for students and trainees new to the technique.  As someone who is very familiar with using Xenopus oocytes to characterize ion transporters, I was pleased to learn about approaches that I was previously not familiar with, such as micro-transplantation and SOC.  Overall, this is a very nice contribution that will be of great use to the field of membrane biology. 

We thank the reviewer for recognizing our effort in writing a sort of guide for researchers and students familiar or not familiar with the methods

 Below are some minor suggestions to improve the manuscript:

 - The addition of a few figures showing representative data from published studies would be helpful for illuminating some of the techniques that are not that easy for a non-expert reader. For example, the description of pre-steady state currents and their interpretation (lines 203-212) are not easily understood.  A figure illustrating the details the authors are referring to along with some representative data would help better communicate this approach.   Figures would also be helpful for comprehension of the VCF technique (lines 231-240) and fluorescence monitoring (lines 299-328). 

A few minor spelling/grammatical errors:

  1. line 233, ‘resides’ should be ‘residues’;     
  2. Line 370,   ‘ATP’ should be ‘ATPase’

We have added the suggested figures, in particular in the PSS figure we have opted for a modified version of an old cartoon published by  H. Lester [1]. This figure does not represent the latest structural model and conformational changes that occur during the transport cycle, but in our opinion, it is a very clear explanation of the pre-steady-state currents and transport currents elicited in the Na+or H+/ coupled solute carriers. For the Voltage clamp fluorimetry, we have reported a simplified version of the technique to be didactical, and finally, for the fluorescence monitoring, we have reported some schematized applications.

Reviewer 2 Report

This is a review of the methodologies uses in Xenopus oocytes to study channels and transporters. The summary of the techniques can be useful but the review in general makes sweeping comments that are not justified and needs significant editing. See below for some specific comments.

The following sentence makes unfounded claims: “Almost all endogenous electrogenic proteins are known [4-12], the multimeric proteins are correctly assembled, and the biophysical parameters are trustable and comparable to that collected in vivo.” It is difficult to claim that ALL endogenous electrogenic protein are known and when they can be activated by exogenous expression. Also the biophysical parameters of channels and transporters are for the most part studied ex vivo no in vivo.

“Xenopus oocytes have a highly efficient biosynthetic apparatus that performs all the post-translational modifications needed for the correct” Again sweeping claims that are difficult to validate.

“the mean life of oocytes is about 10 days” this is incorrect, the maximal oocytes can be kept in the dish is 10 days not the mean life.

Xenopus needs to italicized.

Don’t use positive and negative current but rather inward and outward current.

Line 233: Residues not resides.

For the SOC technique discuss the limitations of fixing and permeabilizing oocytes and the ability to detect the entire protein pool intracellularly.

The paper needs editing for grammar and sentence structure.

Author Response

Reviewer 2

This is a review of the methodologies uses in Xenopus oocytes to study channels and transporters. The summary of the techniques can be useful but the review in general makes sweeping comments that are not justified and needs significant editing. See below for some specific comments.

The following sentence makes unfounded claims: “Almost all endogenous electrogenic proteins are known [4-12], the multimeric proteins are correctly assembled, and the biophysical parameters are trustable and comparable to that collected in vivo.” It is difficult to claim that ALL endogenous electrogenic protein are known and when they can be activated by exogenous expression.

We did not claim that ALL proteins are known but ”almost all” anyway, we have changed the sentence to avoid misunderstanding and added other references about this aspect. In the redline version are highlighted the changes

Also, the biophysical parameters of channels and transporters are for the most part studied ex vivo no in vivo.

“Xenopus oocytes have a highly efficient biosynthetic apparatus that performs all the post-translational modifications needed for the correct” Again sweeping claims that are difficult to validate.

“the mean life of oocytes is about 10 days” this is incorrect, the maximal oocytes can be kept in the dish is 10 days not the mean life.

We have made for all three sentences the changes required by the reviewer to be more correct from the biological point of view and to be much more precise in the description.

Xenopus needs to italicized.

We have italicized the Xenopus and also always added laevis in order to always use the correct way of writing the species

Don’t use positive and negative current but rather inward and outward current.

Thanks for the suggestion, this gives us the opportunity to better clarify the concept of the current direction, consequently we have decided to leave in one point also positive and negative current, adding the explanation as outward and inward in order to help the readers not familiar with the electrophysiological techniques to understand papers where the terms “positive and/or negative currents” are used

Line 233: Residues not resides.

Done

For the SOC technique discuss the limitations of fixing and permeabilizing oocytes and the ability to detect the entire protein pool intracellularly.

We have corrected the text according to also discussing the pro and the cons

The paper needs editing for grammar and sentence structure.

Done

Round 2

Reviewer 2 Report

I thank the authors for addressing my concerns. I would in addition recommend adding the following citation as it shows a concrete example of the usefulness of the oocyte system is studying a particular pathway: Xenopus Oocyte As a Model System to Study Store-Operated Ca(2+) Entry (SOCE). (2016) Front Cell Dev Biol 4:66.

Author Response

Thanks to the reviewer for the positive feedback and for the suggestion of the important citation missing.

We have added it to complete the possible use of XLO and as a source of other possible methodologies.